# Long-Term Glucose Forecasting Using a Physiological Model and Deconvolution of the Continuous Glucose Monitoring Signal

**DOI:** 10.3390/s19194338

**Published:** 2019-10-08

**Authors:** Chengyuan Liu, Josep Vehí, Parizad Avari, Monika Reddy, Nick Oliver, Pantelis Georgiou, Pau Herrero

**Affiliations:** 1Centre for Bio-Inspired Technology, Department of Electrical and Electronic Engineering, Imperial College London, London SW7 2AZ, UK; pantelis@imperial.ac.uk; 2Department of Electrical and Electronic Engineering, Universitat de Girona and with CIBERDEM, Girona 17004, Spain; josep.vehi@udg.edu; 3Department of Medicine, Imperial College Healthcare NHS Trust, London W12 0HS, UK; p.avari@imperial.ac.uk (P.A.); m.reddy@imperial.ac.uk (M.R.); nick.oliver@imperial.ac.uk (N.O.)

**Keywords:** type 1 diabetes, glucose prediction, continuous glucose monitoring, physiological modelling, deconvolution, artificial pancreas

## Abstract

(1) Objective: Blood glucose forecasting in type 1 diabetes (T1D) management is a maturing field with numerous algorithms being published and a few of them having reached the commercialisation stage. However, accurate long-term glucose predictions (e.g., >60 min), which are usually needed in applications such as precision insulin dosing (e.g., an artificial pancreas), still remain a challenge. In this paper, we present a novel glucose forecasting algorithm that is well-suited for long-term prediction horizons. The proposed algorithm is currently being used as the core component of a modular safety system for an insulin dose recommender developed within the EU-funded PEPPER (Patient Empowerment through Predictive PERsonalised decision support) project. (2) Methods: The proposed blood glucose forecasting algorithm is based on a compartmental composite model of glucose–insulin dynamics, which uses a deconvolution technique applied to the continuous glucose monitoring (CGM) signal for state estimation. In addition to commonly employed inputs by glucose forecasting methods (i.e., CGM data, insulin, carbohydrates), the proposed algorithm allows the optional input of meal absorption information to enhance prediction accuracy. Clinical data corresponding to 10 adult subjects with T1D were used for evaluation purposes. In addition, in silico data obtained with a modified version of the UVa-Padova simulator was used to further evaluate the impact of accounting for meal absorption information on prediction accuracy. Finally, a comparison with two well-established glucose forecasting algorithms, the autoregressive exogenous (ARX) model and the latent variable-based statistical (LVX) model, was carried out. (3) Results: For prediction horizons beyond 60 min, the performance of the proposed physiological model-based (PM) algorithm is superior to that of the LVX and ARX algorithms. When comparing the performance of PM against the secondly ranked method (ARX) on a 120 min prediction horizon, the percentage improvement on prediction accuracy measured with the root mean square error, A-region of error grid analysis (EGA), and hypoglycaemia prediction calculated by the Matthews correlation coefficient, was 18.8%, 17.9%, and 80.9%, respectively. Although showing a trend towards improvement, the addition of meal absorption information did not provide clinically significant improvements. (4) Conclusion: The proposed glucose forecasting algorithm is potentially well-suited for T1D management applications which require long-term glucose predictions.

## 1. Introduction

Type 1 diabetes (T1D) is an autoimmune condition characterized by elevated blood glucose levels due to the lack of endogenous insulin production [1]. People with T1D require exogenous insulin delivery to regulate glucose levels. Current therapies for T1D management require measuring capillary glucose levels several times per day and the administration of insulin by means of multiple daily injections (MDI) or continuous subcutaneous insulin infusion (CSII) with pumps. More recently, the improvement on accuracy of subcutaneous continuous glucose monitoring (CGM) has enabled access to virtually continuous glucose concentrations measurements (e.g., every 5 min), glucose trends, and their retrospective analysis. In addition, real-time CGM devices include alerts and alarms for concentrations outside of specified ranges and/or rapid changes in glucose. Clinical data suggest that CGM can improve overall glucose control, as measured by glycated haemoglobin (HbA1c) [2], and can reduce the burden of extreme glucose values (hypo- and hyperglycaemia) [3]. In addition, CGM technology has opened the door to new technologies for managing glucose levels such as sensor-augmented insulin pumps with low-glucose insulin suspension [4] and the artificial pancreas [5]. One additional benefit of the CGM technology is that it facilitates the forecasting of blood glucose levels, and consequently, enables pre-emptive action to avoid undesired events, such as hypoglycaemia and hyperglycaemia.

Glucose forecasting in T1D is a relatively mature field with several algorithms having been proposed, and comprehensive and extensive reviews published, which provides a taxonomy of the different types of existing algorithms [6,7,8]. In particular, four main types of approaches were identified depending on the type of model being used: physiological models [9], data-based models [10,11,12], hybrid models [13], and control relevant models [14,15,16]. Another distinction that can be done among the existing algorithms is based on the type of inputs being considered. A significant proportion of these algorithms use CGM data as the unique source of information to forecast glucose levels [17], while other algorithms use additional exogenous inputs such as meal intake and insulin doses [9], and only a few of them take into account physical exercise [18]. Furthermore, additional information such as meal absorption information could potentially further improve such accuracy [19]. Regarding the prediction horizon, most of the algorithms focus on short-term predictions (i.e., <60 min) [20], with fewer works focusing on longer prediction horizons [9,21,22]. Of note, it has been shown that for short prediction horizons (e.g., 30 min), differences in the forecasting accuracy among different algorithms is marginal [23]. Although short-term prediction horizons have been proven to be useful for applications such as predictive glucose alerts and low-glucose suspension systems [4], they are usually not sufficient for other applications such as closed-loop insulin delivery [5] and insulin dosing decision support [24]. Hence, there is a need for further investigation of this topic.

Finally, there has been a recent increase in the use of machine learning techniques for glucose forecasting [12,22,25]. However, these techniques tend to require a significant amount of data in order to be trained, which might be a limiting factor in clinical practice. A proof of this interest is the recent organisation of the Blood Glucose Level Prediction Challenge as part of the 3rd International Workshop on Knowledge Discovery in Healthcare Data [26], and a recent publication of a literature review [8].

As a result of this research effort, commercial applications have started to appear embedded within sensor-augmented insulin pumps (e.g., Medtronic MiniMed 640G with Smart Guard; Tandem t:slim X2™ with Basal-IQ Technology) which have been proven to reduce nocturnal hypoglycaemia by using predictive glucose alerts and a predictive low-glucose insulin suspension system [27,28].

In this paper, we hypothesise that a glucose prediction algorithm based on a physiological model of glucose–insulin dynamics and a deconvolution technique using CGM data to estimate the model states is a good candidate for performing long-term glucose predictions. In addition, we hypothesise that adding information about meal absorption can enhance prediction accuracy. The presented algorithm is currently being clinically evaluated as part of the safety system of a mobile-based decision support system for T1D management within the framework of the European project PEPPER (Patient Empowerment through Predictive PERsonalised decision support) [29].

## 2. Methods

The proposed glucose prediction algorithm is based on the composite minimal model of glucose–insulin regulation in T1D [30] that uses deconvolution of the CGM signal to estimate some of the model states. In particular, the states of the gastrointestinal model are estimated using the technique proposed by Herrero et al., which has been proven to be an effective way to estimate the glucose rate of appearance from mixed meals [31]. Then, meal information (i.e., carbohydrate amount and absorption type) and insulin boluses are considered as exogenous inputs. Note that, although more complex models are available in the literature [32], the employed composite minimal model was chosen because of its trade-off between simplicity, which facilitates the parameter identification task, and ability to capture the complex glucose–insulin dynamics. The effectiveness of the employed composite model was evaluated by Gillis et al. for glucose prediction using a Kalman filter technique [33] and by Herrero and associates for detecting faults in insulin pump therapy [30].

### 2.1. Composite Minimal Model

The employed composite model of glucose regulation in T1D is composed of the minimal model of glucose disappearance proposed by Bergman and colleagues [34], and the insulin and carbohydrate absorption models proposed by Hovorka et al. [35].

#### 2.1.1. Minimal Model of Glucose Disappearance

The minimal model of glucose disappearance [34] is described by the equations: (1)$$\begin{array}{cc} \dot{G}\left(t\right) & =-\left(S_{G}+X\left(t\right)\right)G\left(t\right)+S_{G}G_{b}+\frac{R_{a}\left(t\right)}{VW}, \end{array}$$(2)$$\begin{array}{cc} \dot{X}\left(t\right) & =-p_{2}X\left(t\right)+p_{2}S_{I}I\left(t\right), \end{array}$$
where G(t) is the glucose concentration at time *t*; X(·) is the insulin action; $R_{a}\left(\cdot \right)$ is the glucose rate of appearance from ingested meals; I(·) is the plasma insulin concentration; $S_{G}$ is the fractional glucose effectiveness; $S_{I}$ is the insulin sensitivity, which models the ratio between endogenous glucose production and glucose uptake; $p_{2}$ is the insulin action rate; $G_{b}$ is the basal glucose; *V* is the distribution volume; and *W* is the subject’s body weight.

#### 2.1.2. Insulin Absorption Model

The plasma insulin concentration is estimated by means of the subcutaneous insulin absorption model proposed by Hovorka et al. [35], which is described by the following equations.
(3)$$\begin{array}{cc} {\dot{S}}_{1}\left(t\right) & =u_{INS}\left(t\right)-\frac{S_{1}\left(t\right)}{t_{maxI}}, \end{array}$$
(4)$$\begin{array}{cc} {\dot{S}}_{2}\left(t\right) & =\frac{S_{1}\left(t\right)-S_{2}\left(t\right)}{t_{maxI}}, \end{array}$$
(5)$$\begin{array}{cc} \dot{I}\left(t\right) & =-k_{e}I\left(t\right)+\frac{S_{2}\left(t\right)}{V_{i}t_{maxI}}, \end{array}$$
where $S_{1}\left(\cdot \right)$ and $S_{2}\left(\cdot \right)$ are the subcutaneous short-acting insulin compartments, I(·) denotes the plasma insulin concentration, the input $u_{INS}\left(\cdot \right)$ represents the subcutaneous insulin infusion, $t_{maxI}$ is the time to maximum insulin absorption, $V_{i}$ is the distribution volume of insulin and $k_{e}$ is the decay rate.

#### 2.1.3. Glucose Absorption Model

The glucose rate of appearance ($R_{a}$) is calculated according to the gastrointestinal absorption model by Hovorka et al. [35], which is represented by the equations: (6)$$\begin{array}{cc} {\dot{R}}_{a1}\left(t\right) & =-\frac{R_{a1}\left(t\right)-A_{g}u_{CHO}\left(t\right)}{t_{maxG}}, \end{array}$$(7)$$\begin{array}{cc} {\dot{R}}_{a}\left(t\right) & =-\frac{R_{a}\left(t\right)-R_{a1}\left(t\right)}{t_{maxG}}, \end{array}$$
where $R_{a1}\left(\cdot \right)$ denotes the glucose appearance in the first compartment, $R_{a}\left(\cdot \right)$ represents the rate of glucose appearance, the model input $u_{CHO}\left(\cdot \right)$ denotes the carbohydrate intake amount, $t_{maxG}$ is the time to maximum glucose rate of appearance and $A_{g}$ is the carbohydrate bioavailability.

### 2.2. Accounting for Meal Absorption

Meal composition has a profound effect on blood glucose levels [19]. Therefore, taking this information into account can potentially enhance glucose estimation and forecasting accuracy. To account for this information in a practical way from the user’s perspective, meals were classified as fast, medium and slow absorption. In particular, fast-absorption meals were considered to have more than 60% of the area under the curve (AUC) of the rate of glucose appearance ($R_{a}$) profile and appeared within the first two hours after the meal ingestion; slow-absorption meals with less than 80% of AUC of the $R_{a}$ profile appeared within four hours, and medium-absorption meals in the middle. To take meal absorption information into account within the employed glucose absorption model, the time-to-maximum absorption rate $t_{maxG}$ is redefined for the duration of each meal as follows:(8)$$\begin{array}{c} {\hat{t}}_{maxG}:=\left\{\begin{array}{cc} t_{maxG}-t_{l}, & u_{ABS}=\mathrm{fast}\;\mathrm{absorption}\\ t_{maxG}, & u_{ABS}=\mathrm{medium}\;\mathrm{absorption}\\ t_{maxG}+t_{d}, & u_{ABS}=\mathrm{slow}\;\mathrm{absorption} \end{array}\right. \end{array}$$
where ${\hat{t}}_{maxG}$ is the time-to-maximum absorption rate accounting for the meal absorption; $t_{l}$ and $t_{d}$ represent the time shift on the time-to-maximum absorption rate due to different meal absorption rates; and $u_{ABS}$ is the absorption type of the meal (fast, medium and slow). In particular, $t_{l}$ and $t_{d}$ were fixed to 20 min based on experiments carried out using the UVa-Padova simulator [36]. Figure 1 shows the average $R_{a}$ profiles corresponding to the fast, slow and medium meals, as per definition above, from the UVa-Padova simulator for a 60 g intake of carbohydrates.

### 2.3. Glucose Prediction Algorithm

The proposed glucose prediction algorithm uses a discretised version of the presented composite model (Equations (Equation 1)–(Equation 7)). For this purpose, a forward Euler’s configuration with 1 min step size is used to simulate the model.

Let:(9)$$\begin{array}{c} x(k)=f(x(k-1),p,u(k-1)), \end{array}$$
be the system equations representing a discretised version of the described composite model, where *k* denotes the sampling instant. Let $x=\left[\begin{array}{ccccccc} G\; & X\; & S_{1}\; & S_{2}\; & I\; & R_{a1}\; & R_{a} \end{array}\right]$ represent the model states; $p=\left[\begin{array}{cccccccccc} k_{e}\; & \;t_{maxI}\; & \;V_{i}\; & \;A_{g}\; & \;t_{maxG}\; & \;S_{G}\; & \;p_{2}\; & \;W\; & \;V\; & \;S_{I} \end{array}\right]$ represent the model parameters; and $u\;=\;\left[\begin{array}{ccc} u_{CHO}\; & \;u_{INS}\; & \;u_{ABS} \end{array}\right]$ represent the model inputs, where $u_{CHO}$ denotes the amount of ingested carbohydrates, $u_{INS}$ denotes the insulin boluses (units), and $u_{ABS}$ denotes the meal absorption type (slow, medium, fast).

First, the model states are initialised as $x\left(k-1\right)=\left[\begin{array}{ccccccc} G_{CGM}\left(\cdot \right)\; & 0\; & 0\; & 0\; & 0\; & 0\; & 0 \end{array}\right]$, being $G_{CGM}\left(\cdot \right)$ the current CGM measurement. Then, Equation (Equation 9) is evaluated *M* times, *M* being the number of minutes elapsed between two consecutive CGM measurements (e.g., 5 min → five times). In parallel, the model states of the gastrointestinal model (Equations (Equation 6) and (Equation 7)) are estimated in real-time by applying a deconvolution of the CGM signal using the technique proposed by Herrero et al. [31]. In particular, the glucose rate of appearance (${\hat{R}}_{a}\left(\cdot \right)$) in the second compartment is estimated as:(10)$$\begin{array}{c} {\hat{R}}_{a}\left(k\right)=\left[G_{CGM}^{\prime }\left(k\right)+\left(S_{G}+X\left(k\right)\right)G_{CGM}\left(k\right)-S_{G}G_{b}\right]VW, \end{array}$$
where $G_{CGM}^{\prime }\left(\cdot \right)$ is the derivative of the glucose measurements calculated as the slope of the linear regression of three consecutive glucose values, and X(·) is the insulin action calculated with Equation (Equation 2). In order to reduce the influence of the measurement disturbance, the derivative is bounded by $|{\bar{G}}^{\prime }\left(\cdot \right)|\leq 1$ mg/dL per min. To further reduce the effect of sensor noise on ${\hat{R}}_{a}\left(\cdot \right)$, a moving average filter is applied to the signal:(11)$$\begin{array}{c} {\tilde{R}}_{a}\left(k\right):=\frac{\sum_{i=k-n+1}^{i=k-1}{\tilde{R}}_{a}\left(i\right)+{\hat{R}}_{a}\left(k\right)}{n}, \end{array}$$
where ${\tilde{R}}_{a}\left(\cdot \right)$ is the filtered signal and *n* is the length of the moving window (n=3).

Then, the glucose appearance in the first compartment is estimated by:(12)$$\begin{array}{c} {\tilde{R}}_{a1}\left(k\right)={\tilde{R}}_{a}^{\prime }\left(k\right)t_{maxG}+{\tilde{R}}_{a}\left(k\right), \end{array}$$
where ${\tilde{R}}_{a}^{\prime }\left(\cdot \right)$ is the derivative of ${\tilde{R}}_{a}\left(\cdot \right)$ approximated by calculating the slope of the linear regression of three consecutive values.

Then, the estimated model states being used for forecasting purposes are calculated by means of a weighted average between the calculated states with the discretised model ($R_{a}$ and $R_{a1}$) and the states estimated by deconvolution (${\tilde{R}}_{a}$ and ${\tilde{R}}_{a1}$) as follows:(13)$$\begin{array}{cc} {\overset{ˇ}{R}}_{a}\left(k\right) & :=Q_{1}{\tilde{R}}_{a}\left(k\right)+\left(1-Q_{1}\right)R_{a}\left(k\right), \end{array}$$(14)$$\begin{array}{cc} {\overset{ˇ}{R}}_{a1}\left(k\right) & :=Q_{1}{\tilde{R}}_{a1}\left(k\right)+\left(1-Q_{1}\right)R_{a1}\left(k\right). \end{array}$$
where $Q_{1}\in \left[0,1\right]$ is a tuning parameter that allows putting more weight on the model estimation or on the estimation by the deconvolution technique.

Similarly, the blood glucose state being used for forecasting purposes is calculated as:(15)$$\begin{array}{c} \overset{ˇ}{G}\left(k\right):=Q_{2}G_{CGM}\left(k\right)+\left(1-Q_{2}\right)G\left(k\right), \end{array}$$
where $Q_{2}\in \left[0,1\right]$ is a tuning parameter that allows putting more weight on the model estimation (G(·)) or on the CGM measurement ($G_{CGM}\left(\cdot \right)$). Note that $Q_{1}$ and $Q_{2}$ are constant. In particular, we used $Q_{1}=Q_{2}=0.5$ when identifying the model parameters and $Q_{1}=Q_{2}=0.7$ when testing the model since these were the values leading to better population results.

Finally, the model states are updated as:(16)$$\begin{array}{c} x\left(k\right)=\left[\begin{array}{ccccccc} \overset{ˇ}{G}\left(k\right)\; & X(k)\; & S_{1}\left(k\right)\; & S_{2}\left(k\right)\; & I(k)\; & {\overset{ˇ}{R}}_{a1}\left(k\right)\; & {\overset{ˇ}{R}}_{a}\left(k\right) \end{array}\right], \end{array}$$
and the model is valuated over the predefined prediction horizon (PH) to obtain the forecasted states x(k+PH), where G(k+PH) is the desired forecasted glucose value. Figure 2 shows a graphical representation of the described glucose forecasting algorithm.

### 2.4. Model Parameter Identification

The proposed prediction algorithm can be individualised by identifying some of the model parameters by using retrospective data. Since identification of all model parameters is difficult due to identifiability problems, some of the parameters, which are known to have less inter-subject variability, were fixed to mean populations values (i.e., $S_{G}$, *V*, $V_{i}$, $k_{e}$, $p_{2}$, $A_{g}$) [37], while others were set by using a priori known information from the subjects, such as body weight (*W*) and basal glucose ($G_{b}$). Finally, parameters $S_{I}$, $t_{maxI}$ and $t_{maxG}$ were identified by using an optimisation technique that minimises the mean absolute relative difference (MARD) between the predicted glucose (G(k+PH)) and the corresponding glucose measurements ($G_{CGM}\left(k+PH\right)$). Matlab fmincon constrained optimisation routine was employed for this purpose. Constraints for the identified parameters were $S_{I}\in \left[0.001,0.005\right]$ min${}^{-1}$ per μU/mL, $t_{maxI}\in \left[50,140\right]$ min and $t_{maxG}\in \left[50,140\right]$ min. Note that parameters $S_{I}$, $t_{maxI}$ and $t_{maxG}$ were identified for each one of the evaluated prediction horizons. Table 1 shows the employed values for the model parameters indicating which ones are a priori known and which ones are identified.

### 2.5. Baseline Algorithms

In order to compare the performance of the proposed algorithm, validated and commonly employed glucose forecasting algorithms from the literature were chosen. In particular, a third-order auto regression with exogenous input (ARX) method was chosen because it has shown good performance when compared to other algorithms and it is easy to implement [23]. Then, a latent variable with exogenous input (LVX) method was selected because it has showed superiority when compared against other existing techniques and its source code is publicly available [38]. Of note, more recent algorithms, such as [9,21], could have also been chosen, but their complexity of implementation and difficulty in replicating the reported results was the main reason for not doing so.

### 2.6. Clinical Data Testing

To test the proposed algorithm, a two-week clinical dataset obtained from 10 adult subjects with T1D undergoing a clinical trial carried out at Imperial College Healthcare NHS Trust, London, UK evaluating the benefits of an advanced insulin bolus calculator was employed [39]. In particular, one week worth of data per participant was used for parameter identification purposes (mean ± std: 1897±143 data points) (physiological model-based (PM) algorithm), while the other week was employed for testing purposes (mean ± std: 1909±124 data points). Since no reliable information about meal composition was available, the assumption that breakfast and snacks were fast absorption meals, and lunch and dinner were medium absorption meals, was made. Note that not having such information might limit the benefits of accounting for meal absorption information. In an ideal scenario, CGM measurements are received every 5 min. However, it is common to have gaps in the data. In order to minimise the impact of such gaps in the proposed method, a modified Akima cubic Hermite interpolation method was employed for data imputation purposes.

### 2.7. In Silico Testing

In order to evaluate the potential benefit of using meal absorption within the proposed algorithm, a modified version of the UVa-Padova T1DM simulator (v3.2) [36] was employed. Intra-day variability was emulated by modifying some of the parameters of the model described in [32]. In particular, meal variability was emulated by introducing meal-size variability (CV=10%), meal-time variability (STD=20) and uncertainty in the carbohydrate estimation (uniform distribution between −30% and +20%) [40]. Meal absorption rate ($k_{abs}$) and carbohydrate bioavailability (*f*) were considered to vary by ±30% and ±10% respectively. To account for variability in meal composition, the 33 available meals in the simulator were considered. Note that each cohort (adults, adolescent and children) has 11 different meals. In addition, 16 new meals obtained by using the technique for estimating the rate of glucose appearance proposed by Herrero et al. in [31] were included. Details about the mixed meal library are provided in Appendix A. By using the absorption classification criteria introduced in Section 2.2, of the 49 considered meals, 31 were classified as fast absorption, 15 as medium absorption and three as slow absorption. Finally, intra-subject variability in insulin absorption model parameter ($k_{d}$, $k_{a1}$, $k_{a2}$) was assumed as ±30% [41,42]. The 10 available adult subjects were used for this purpose. The open-loop insulin therapy provided by the simulator was employed to generate the datasets. A two-week scenario with a daily pattern of carbohydrate dose intake of 7 am (±20 min) (70 g), 13 pm (±20 min) (100 g) and 7 pm (±20 min) (80 g) was chosen. The first week of data was used for the parameter estimation (2016 data points), while the second week works for testing purposes (2016 data points).

### 2.8. Evaluation Metrics

To measure the forecasting accuracy of the evaluated algorithms, the root mean square error (RMSE), and error grid analysis (EGA) [43], were used. RMSE is calculated as:(17)$$\begin{array}{c} RMSE=\sqrt{\frac{\sum_{k=1}^{N-PH}{\left(G\left(k+PH\right)-G_{CGM}\left(k+PH\right)\right)}^{2}}{N-PH}}, \end{array}$$
where G(k+PH) is the forecasted glucose value as indicated in Section 2.3, $G_{CGM}\left(k+PH\right)$ is the corresponding CGM measurement, and *N* is the total number of glucose data pairs.

EGA is employed to represent the clinical significance of the error between the forecasted glucose value and the actual measurement. In particular, the A region of EGA corresponds to accurate glucose predictions, meaning that forecasted glucose values deviates less than ±20% from the actual CGM measurements (i.e., $|G\left(k+PH\right)-G_{CGM}\left(k+PH\right)|\leq 20\%\;G_{CGM}\left(k+PH\right)$), or when both the forecasted and the actual measurements indicate hypoglycaemia (i.e., G(k+PH)≤70 mg/dL and $G_{CGM}\left(k+PH\right)\leq 70$ mg/dL). The B region corresponds to forecasted values with a deviation from the CGM measurements larger than 20% (i.e., $|G\left(k+PH\right)-G_{CGM}\left(k+PH\right)|\geq 20\;\%\;G_{CGM}\left(k+PH\right)$) but would not lead to inappropriate treatment (see regions C,D,E). The C region represents forecasted glucose values are more than 100 mg/dL below the actual measurement (i.e., $G\left(k+PH\right)\leq G_{CGM}\left(k+PH\right)-100$ mg/dL), or forecasted glucose is indicating hypoglycaemia while the CGM measurements is between 130 and 180 mg/dL (i.e., G(k+PH)≤70 mg/dL and 130 mg/dL $\leq G_{CGM}\left(k+PH\right)\geq 180$ mg/dL). Finally, the D region represents the zone with forecasted glucose values in the target range while the actual measurement is out of the target range (i.e., 70 mg/dL ≤G(k+PH)≥180 mg/dL and $G_{CGM}\left(k+PH\right)\leq 70$ mg/dL $||\;G_{CGM}\left(k+PH\right)\geq 180$ mg/dL). The E region corresponds to potentially clinically dangerous glucose predictions, meaning forecasted glucose indicating hypoglycaemia while CGM measurements indicating hyperglycaemia (i.e., G(k+PH)≤70 mg/dL and $G_{CGM}\left(k+PH\right)\geq 180$ mg/dL), or when forecasted glucose indicate hyperglycaemia while CGM measurements indicate hypoglycaemia (i.e., G(k+PH)≥180 mg/dL and $G_{CGM}\left(k+PH\right)\leq 70$ mg/dL).

In addition, the efficiency of the algorithm at predicting hypoglycaemia was evaluated by the Matthews correlation coefficient (MCC), defined as:(18)$$\begin{array}{c} MCC\;=\;\frac{TP\cdot TN\;-\;FP\cdot FN}{\sqrt{\left(TP\;+\;FP)(TP\;+\;FN)(TN\;+\;FP)(TN\;+\;FN\right)}}, \end{array}$$
where TP denotes the number of true positives, i.e., predictions of hypoglycemic events that are confirmed to be actual episodes of hypoglycemia, with a hypoglycemic event being defined as three consecutive glucose values below 70 mg/dL; FN denotes the number of false negatives, i.e., missed predictions of hypoglycemic events; TN denotes the number of true negatives, i.e., correct prediction of glucose values above 70 mg/dL; and FP denotes the number of false positives, i.e., false predictions of hypoglycemic events.

Assessment of statistical significance for between-method differences was performed, with 0.1%, 1% and 5% confidence levels, using a paired *t*-test as implemented in Matlab.

Finally, prediction horizons of 30, 60, 90, and 120 min were employed to compare the studied algorithms.

## 3. Results

### 3.1. Clinical Data Results

Baseline characteristics of the 10 selected participants (one male and nine female) had a median (interquartile range (IQR)) age of 29.5.5 (25.0–42.7) years, duration of diabetes 16.0 (9.2–24.7) years, BMI 26.0 (24.4–31.0) Kg/m${}^{2}$, and HbA1c 53.0 (51.0–59.7) mmol/mol.

The distribution of the identified model parameters for the selected 10-adult cohort is $S_{I}=0.0033\pm 0.0015$ min${}^{-1}$ per μU/mL, $t_{maxI}=78.36\pm 16.52$ min and $t_{maxG}=85.23\pm 24.86$ min.

Table 2 shows the results corresponding to the 10 adult cohort for baseline algorithms (LVX and ARX) and the proposed method, without accounting for meal absorption information (PM) and accounting for it ($PM_{MA}$). In order to show the consistency of the results at the individual level, the RMSE results for each of the evaluated subjects are presented in Appendix B (Table A4). The normality of the reported data distributions was evaluated by means of the Shapiro–Wilk test and, in all cases, the null hypothesis for an α=0.05 was not rejected.

Figure 3 depicts the percentage of improvement, for each of the prediction horizons, of three of the evaluated metrics (RMSE, A-region of EGA, and MCC) when comparing the proposed method versus the ARX model, which ranks second in terms of prediction accuracy.

Figure 4 shows a 24 h period close up for a selected individual showing the prediction results for the three evaluated forecasting methods (LVX, ARX, and PM) with a prediction horizon of 120 min.

### 3.2. In Silico Results

The distributions of the identified model parameters for the selected in silico cohort of 10 adults with T1D, expressed as Mean±STD, are: $S_{I}=0.00275\pm 0.0014$ min${}^{-1}$ per μU/mL, $t_{maxI}=114.6\pm 21.6$ min and $t_{maxG}=68.9\pm 6.8$ min.

Table 3 presents the results corresponding to the 10-adult cohort for each one of the selected baseline algorithms (LVX and ARX) and the proposed method, without accounting for meal absorption information (PM), and accounting for it (PM${}_{MA}$).

## 4. Discussion

Results obtained with both clinical data (Table 2) and in silico data (Table 3) are consistent at showing that, when compared to commonly employed glucose forecasting algorithms (LVX and ARX), the proposed PM algorithm is superior on all evaluated prediction horizons. As observed in Figure 3, such superiority is more evident at longer prediction horizons, and in particular, when looking at the MCC. In the context of predictive glucose alarms, this improvement would be translated into a system with less false alarms and missed alarms, while in an automatic insulin delivering system, it could be translated into better glucose control.

This improvement in forecasting accuracy at longer horizons might be attributable to the use of a physiological model, which is able to better account for the long-term glucose–insulin dynamics when compared to purely data-driven based approaches.

As observed in Table 2 and Table 3, the inclusion of meal absorption information into the PM${}_{MA}$ method did not yield significant improvements in the forecasting accuracy. Although additional meals were included into the UVA-Padova simulator to increase its inter-meal variability, such variability might still not be enough to show the benefits of the proposed approach. Note that the presented mixed-meal model library in Appendix A is per se a scientific outcome that can be used in other applications [44].

It is worth noting that, when comparing against the baseline algorithms, the proposed method shows better performance on the real cohort than on the in silico. Our hypothesis to explain this fact is that our method is better at capturing the complexity of the real world glucose–insulin dynamics as a result of using a deconvolution technique for state estimation.

Although the proposed approach has been proven to be superior to some of existing forecasting algorithms, there are still some scenarios were there is a significant mismatch between the predicted glucose and the actual glucose, e.g., Figure 4 around 15:00 h. This might be explained by the presence of an unaccounted perturbation such as physical exercise. Note that for this particular scenario, a closed-loop control algorithm, such as Model Predictive Control, could lead to sub-optimal results, but this is a limitation inherent to this type of feedback controller.

Finally, unlike recently proposed machine learning methods, which require a significant amount of data to be trained, the proposed methods requires a relatively small dataset (e.g., one week), which makes it practical for clinical applications.

## 5. Conclusions and Future Work

The proposed glucose forecasting algorithm based on a physiological model and a state estimation deconvolution technique is a potential solution for diabetes technology solutions that require long-term glucose predictions (e.g., 120 min), such as closed-loop insulin delivery and insulin dosing decision support. Future work to further improve the accuracy of the proposed algorithm includes accounting for circadian variations on insulin sensitivity [45] and the short- and long-term effect of exercise on insulin requirements [46]. Finally, online parameter estimation is another potential avenue for investigation.

## Figures and Tables

**Figure 1 sensors-19-04338-f001:**
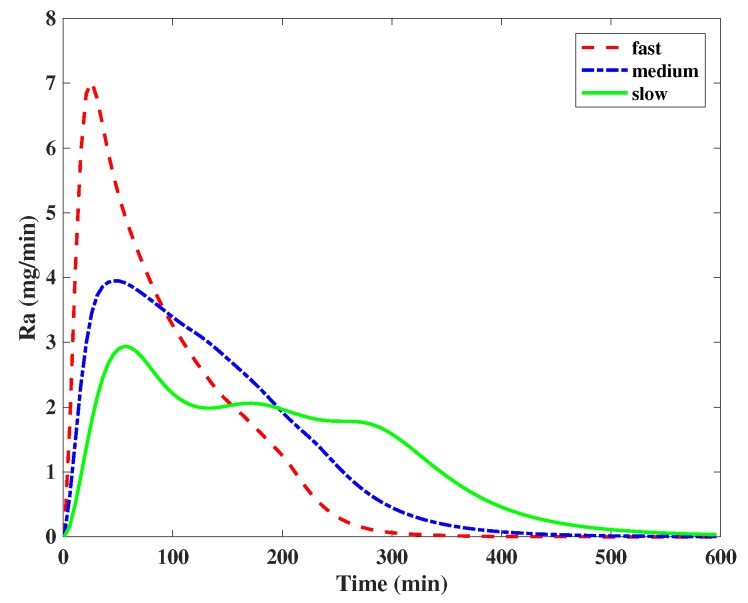
Average $R_{a}$ profiles corresponding to the fast, slow and medium meals from the UVa-Padova simulator for a 60 g intake of carbohydrates.

**Figure 2 sensors-19-04338-f002:**
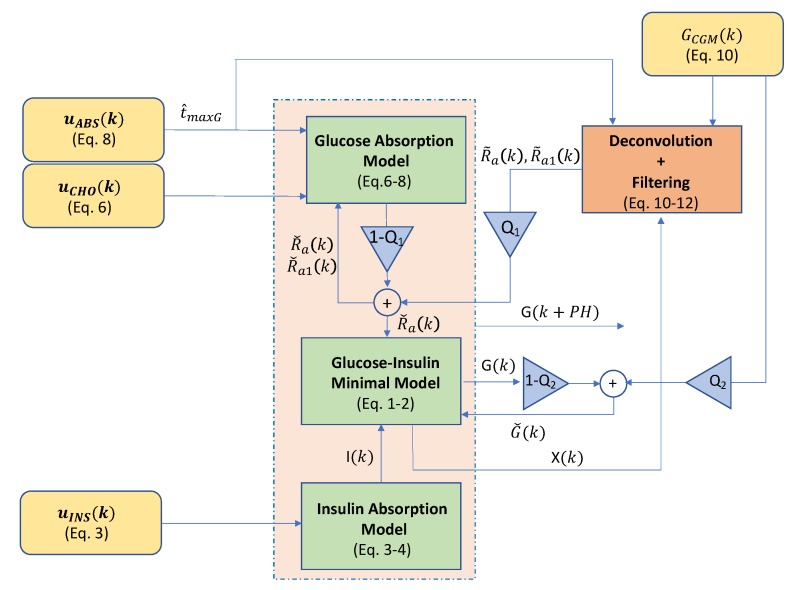
Block diagram corresponding to the proposed glucose forecasting algorithm. The whole diagram is executed every time a glucose value ($G_{CGM}$) (continuous glucose monitoring (CGM)) is received. Then, the physiological model represented by the green blocks is evaluated over the prediction horizon (PH) to obtain the forecasted glucose (G(k+PH)).

**Figure 3 sensors-19-04338-f003:**
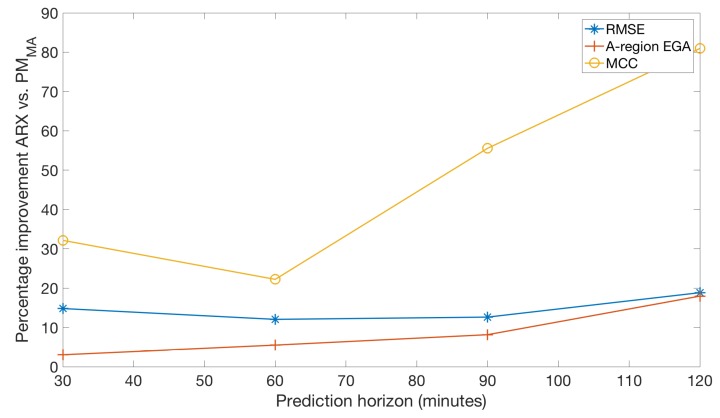
Average percentage of improvement against prediction horizon for three of the evaluated metrics (RMSE, A-region of EGA, and MCC) when comparing the proposed method $PM_{MA}$ versus the ARX model on the 10-adult real cohort.

**Figure 4 sensors-19-04338-f004:**
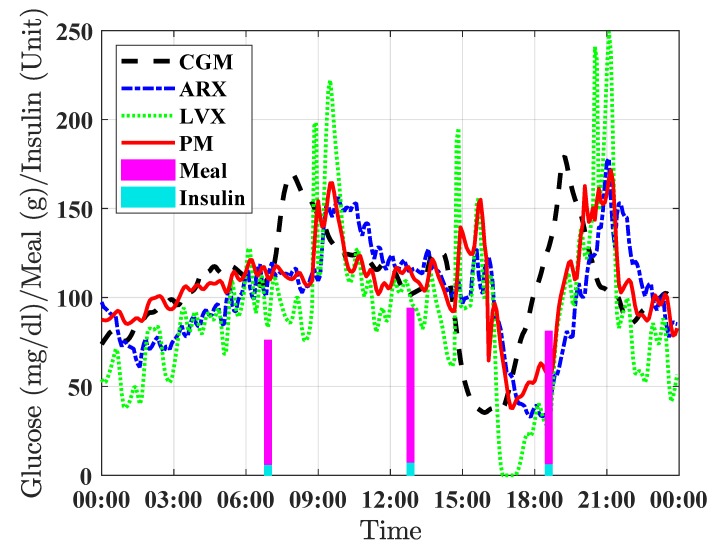
Example of 24 h period close up for a representative real individual showing the prediction results for the three evaluated forecasting methods with a prediction horizon of 120 min. The continuous glucose measurements is represented by the dashed black line, the prediction by the proposed PM method is displayed in solid-red line, results for the LVX and ARX methods are showed in dotted green line and dash-dotted blue line respectively. Vertical pink bars indicate carbohydrate intakes (grams) and vertical light blue bars indicate insulin boluses (units).

**Table 1 sensors-19-04338-t001:** Values of the parameters used in the forecasting algorithm. * indicates parameters that are identified and ∗∗ indicates parameters that are known from a priori information from the subjects. The rest of the parameters are fixed to mean population values obtained from the scientific literature [31,35].

Parameter	$S_{G}$	$S_{I}$	$G_{b}$	*V*	$V_{i}$	*W*	$t_{\mathrm{maxI}}$
Value	0.02	*	∗∗	0.9	1.2	∗∗	*
Units	$\min^{-1}$	$\min^{-1}\;\mathrm{per}\;\mu U/\mathrm{mL}$	mg/dL	$\mathrm{dL}/{\mathrm{kg}}^{2}$	mL/kg	kg	min
Parameter	$t_{maxG}$	$k_{e}$	$p_{2}$	$A_{g}$	$Q_{1}$	$Q_{2}$	PH
Value	*	1.5	0.02	0.85	0.5[0.7]	0.5[0.7]	30
Units	min	$\min^{-1}$	$\min^{-1}$	–	–	–	min

**Table 2 sensors-19-04338-t002:** Results corresponding to the 10-adult population expressed as Mean±STD for the studies algorithms and different PH in minutes. Assessment of statistical significance between adjacent rows is indicated with ${}^{*}$ for p<0.001, ${}^{+}$ for p<0.01, and ${}^{‡}$ for p<0.05.

Config-uration	PH =30						
	RMSE (mg/dL)	EGA-regions (%)					MCC
		A	B	C	D	E	
LVX	20.74	85.56	13.25	$0.08^{}$ ^+^	$1.08^{}$ ^+^	0.03	$0.71^{}$ *
	±2.51	±4.55	±4.02	±0.06	±0.71	±0.02	±0.12
ARX	$20.74^{}$ *	85.34	$13.22^{}$ ^+^	$0.06^{}$ ^‡^	1.36	$0.02^{}$ ^‡^	0.67
	±2.65	±4.49	±3.80	±0.05	±0.93	±0.02	±0.12
PM	$19.03^{}$ *	86.21 *	12.65 *	$0.03^{}$ ^‡^	$1.10^{}$ ^+^	0.01	$0.72^{}$ *
	±2.28	±5.07	±3.73	±0.03	±0.94	±0.01	±0.12
$PM_{MA}$	17.67	87.94	11.11	0.02	0.92	0.01	0.74
	±2.12	±4.61	±3.39	±0.02	0.77	±0.01	±0.11
Config-uration	PH =60						
	RMSE (mg/dL)	EGA-regions (%)					MCC
		A	B	C	D	E	
LVX	34.93	67.62	29.26	0.45	$2.47^{}$ ^‡^	$0.20^{}$ *	$0.42^{}$ ^‡^
	±4.85	±6.30	±5.08	±0.27	±1.53	±0.13	±0.14
ARX	$34.52^{}$ ^‡^	67.11	29.39	$0.34^{}$ ^‡^	3.00	$0.16^{}$ ^‡^	0.39
	±4.82	±6.35	±4.76	±0.22	±2.15	±0.12	±0.15
PM	32.69 *	$68.10^{}$ *	$28.70^{}$ *	$0.20^{}$ ^+^	$2.90^{}$ *	$0.10^{}$ ^+^	$0.43^{}$ ^+^
	±4.17	±8.14	±5.64	±0.15	±2.58	±0.08	±0.12
$PM_{MA}$	30.36	70.80	26.32	0.12	2.70	0.06	0.44
	±3.88	±8.05	±5.59	±0.10	±2.48	±0.06	±0.12
Config-uration	PH =90						
	RMSE (mg/dL)	EGA-regions (%)					MCC
		A	B	C	D	E	
LVX	44.56	56.36	38.57	1.04	3.49	0.54	$0.30^{}$ ^‡^
	±7.76	±6.68	±4.66	±0.65	±2.06	±0.46	±0.14
ARX	$43.65^{}$ ^‡^	56.45	38.25	0.87	4.00	$0.43^{}$ ^‡^	$0.27^{}$ ^+^
	±7.24	±6.74	±4.54	±0.65	±2.77	±0.33	±0.14
PM	$41.07^{}$ *	$58.17^{}$ *	$37.09^{}$ *	$0.53^{}$ *	$3.91^{}$ ^‡^	$0.30^{}$ ^+^	$0.38^{}$ *
	±5.65	±7.63	±4.96	±0.25	±2.94	±0.21	±0.10
$PM_{MA}$	38.14	61.05	34.73	0.33	3.70	0.19	0.42
	±5.25	±7.69	±4.94	±0.18	±3.91	±0.13	±0.11
Config-uration	PH =120						
	RMSE (mg/dL)	EGA-regions (%)					MCC
		A	B	C	D	E	
LVX	51.30	48.99	44.22	1.53	4.23	1.03	$0.22^{}$ ^‡^
	±10.26	±6.83	±4.31	±1.05	±2.35	±0.97	±0.12
ARX	$49.85^{}$ *	$49.34^{}$ *	$43.89^{}$ *	$1.31^{}$ *	$4.70^{}$ ^‡^	$0.76^{}$ ^+^	$0.21^{}$ *
	±9.33	±6.72	±4.32	±1.07	±2.95	±0.60	±0.12
PM	$43.57^{}$ *	$55.25^{}$ *	$39.45^{}$ *	$0.69^{}$ *	$4.23^{}$ ^‡^	$0.38^{}$ ^+^	$0.35^{}$ *
	±6.53	±7.46	±4.69	±0.35	±3.04	±0.27	±0.09
$PM_{MA}$	40.46	58.19	37.08	0.45	4.04	0.24	0.38
	±6.06	±7.58	±4.78	±0.25	±3.01	±0.17	±0.10

**Table 3 sensors-19-04338-t003:** Results corresponding to the 10-adult virtual population expressed as Mean±STD for the studies algorithms and different PH in minutes. Assessment of statistical significance between adjacent rows is indicated with ${}^{*}$ for p<0.001, ${}^{+}$ for p<0.01 and ${}^{‡}$ for p<0.05.

Config-uration	PH =30						
	RMSE (mg/dL)	EGA-regions (%)					MCC
		A	B	C	D	E	
LVX	12.85	$94.93^{}$ *	$4.62^{}$ *	0	$0.45^{}$ *	0	0.94
	±1.47	±0.90	±0.81	±0	±0.31	±0	±0.03
ARX	$12.84^{}$ *	$93.80^{}$ ^+^	$5.39^{}$ *	0.01	$0.80^{}$ *	0	$0.95^{2}$
	±1.31	±1.08	±0.96	±0.02	±0.34	±0	±0.02
PM	$10.90^{}$ *	$94.82^{}$ *	$4.61^{}$ *	0	$0.57^{}$ *	0	$0.98^{}$ *
	±1.23	±1.19	±0.79	±0	±0.15	±0	±0.06
$PM_{MA}$	10.06	95.70	3.92	0	0.38	0	0.99
	±1.14	±1.16	±0.82	±0	±0.21	±0	±0.06
Config-uration	PH =60						
	RMSE (mg/dL)	EGA-regions (%)					MCC
		A	B	C	D	E	
LVX	$28.00^{}$ ^‡^	74.07	$24.26^{}$ ^‡^	0.25	$1.41^{}$ *	0.01	$0.79^{}$ ^+^
	±3.62	±4.87	±5.19	±0.37	±0.70	±0.02	±0.08
ARX	$26.38^{}$ ^+^	75.91	20.58	0.07	3.43	0.01	$0.69^{}$ *
	±3.32	±3.84	±3.54	±0.11	±1.02	±0.02	±0.12
PM	$24.44^{}$ *	$76.46^{}$ *	$20.29^{}$ *	0	$3.25^{}$ *	0	$0.83^{}$ *
	±2.61	±4.24	±3.37	±0	±0.61	±0	±0.12
$PM_{MA}$	22.56	79.13	18.02	0	2.85	0	0.87
	±2.41	±4.00	±3.17	±0	±0.49	±0	±0.12
Config-uration	PH =90						
	RMSE (mg/dL)	EGA-regions (%)					MCC
		A	B	C	D	E	
LVX	$41.04^{}$ ^‡^	$55.42^{}$ ^‡^	$40.92^{}$ ^+^	$1.34^{}$ ^+^	$2.20^{}$ *	0.12	$0.53^{}$ *
	±6.21	±11.35	±11.93	±0.75	±1.05	±0.10	±0.13
ARX	$36.10^{}$ ^+^	64.24	29.85	$0.34^{}$ ^‡^	5.35	0.22	$0.47^{}$ *
	±4.92	±5.15	±4.83	±0.33	±1.50	±0.30	±0.15
PM	$33.50^{}$ *	$64.86^{}$ *	$29.97^{}$ *	0.10	4.99 *	$0.08^{}$ ^‡^	$0.55^{}$ *
	±3.51	±5.49	±5.38	±0.16	±0.98	±0.08	±0.16
$PM_{MA}$	30.93	67.75	27.39	0.05	4.79	0.02	0.58
	±3.51	±5.36	±5.17	±0.10	±0.87	±0.05	±0.16
Config-uration	PH =120						
	RMSE (mg/dL)	EGA-regions (%)					MCC
		A	B	C	D	E	
LVX	48.50	46.61	$48.63^{}$ ^‡^	$2.00^{}$ *	$2.41^{}$ *	0.35	$0.38^{}$ *
	±8.97	±14.29	±14.69	±0.82	±1.37	±0.33	±0.16
ARX	$43.03^{}$ *	$55.67^{}$ ^+^	$36.13^{}$ ^‡^	$0.75^{}$ ^+^	$6.80^{}$ ^+^	$0.65^{}$ ^‡^	$0.29^{}$ *
	±6.06	±6.07	±5.34	±0.46	±1.97	±0.62	±0.15
PM	$37.63^{}$ *	$60.72^{}$ *	$33.09^{}$ *	$0.17^{}$ ^‡^	5.87	$0.15^{}$ ^‡^	$0.45^{}$ *
	±4.70	±6.62	$54.01^{}$ *	$54.01^{}$ *	$54.01^{}$ *	$54.01^{}$ *	±0.15
$PM_{MA}$	34.74	63.64	30.52	0.07	5.69	0.07	0.51
	±4.34	±5.52	±5.42	±0.16	±1.02	±0.10	±0.17

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
