# Peer review of "Long-Term Glucose Forecasting Using a Physiological Model and Deconvolution of the Continuous Glucose Monitoring Signal"

_sensors, 2019, doi:10.3390/s19194338_

Round 1

Reviewer 1 Report

See attached file.

Reviewer 2 Report

Paper Format (FF)

The paper is well-organized, but several typos should be fixed to improve legibility:

* PF1: line 67: missing reference.
* PF2: line 152: extra “a”
* PF3: Once defined T1D, why do you not use the acronym everywhere? e.g., in line 93.
* PF4: Which criteria have you followed to sort the references? You have used neither in order of appearance nor last name of the first author.
* PF5: The first author in [30] and [31] is the same; use the same format to identify her.
* From page 5 and on, you start using different symbols (i.e., \bar{G}, \hat{R}_a, and \hat{G}). I miss a distinctive description of their meanings. Hence:
   * PF6: \bar{G} is the sensor measurement which does not coincide with model glucose (G), and, \hat{R}_a is an estimation of the parameter using deconvolution, and its value is different from model values. If so, I think it would be clarifying to state it explicitly.
   * PF7: May you use a different symbol to distinguish between model simulated values (i.e., R_a) and parameter values after applying Eq.(12), Eq.(13), and Eq(14)? Using the same symbols is confusing.
* PF8: According to the International System of Units:
* PF8: there should be a space between the numerical value and unit symbol. Review all along with the paper, specifically Section 2.7 and 2.8.
* PF9: In Fig. 2, the text “\hat{R}_a” should be before Q.
* PF10: You should include the text “\hat{R}_{a1}” along with “\hat{R}_a.” Both are the outputs of the Deconvolution box. And similarly at the output of the box “Glucose Absorption model.”
* PF11: The symbol G_{forecasted} is not introduced in the paper. What is it?
* PF12: I do not see the relationship between the boxes “Meal” and “R_a Deconvolution” in the equations, but you have included an arrow connecting both boxes. To avoid the confusion, you could explicitly illustrate this relationship in the figure by tagging the arrow with the appropriate magnitude.
* PF13: Similarly to the previous bullet, should the tag “I_p(k),” that links the boxes “Insulin Absorption Model” and “R_a Deconvolution” be substituted by “X(k)”?
* PF14: You should include a new box “Glucose disappearance” that calculates “X(k)” and in this way you state (i) that the deconvolution techniques do not depend on any way of the Insulin Absorption Model, and (ii) it only depends on Eq.(2), that only depends on I(t) but does not depend on any other magnitude estimated by the Composite model.
* PF15: In Fig. 2, it would facilitate the reading to include the equation numbers that are computed in each box. E.g., include the labels Eq. (9), (10), and (11) in the box with the text “R_a Deconvolution.”
Also in Fig. 2, the text "\hat{R}_a" should be placed before Q.

Paper Contents (PC)

I have some concerns:

PC1: In Section 2.8, you introduce the symbol \hat{G}, but you do not explicitly present the equation to calculate it. You should. PC2: In Eq. (10), the value of \dot{G} is calculated from the linear regression of \hat{G}, but \dot{G} is previously calculated in Eq. (1). So, I recommend to use different symbols for different magnitudes, and, in this particular case, use \dot{\hat{G}} to represent this magnitude. Regarding the parameter Q:
   * PC3: You use the same Q for all the equations. That means that you have the same trust in model estimation or deconvolution technique for all the magnitudes, either R_{a1} (that is calculated using a cascade of equations (Eq. (9), (10), and (11) with their uncertainties in the parameter’s values) or G (that only uses Eq. (14)). Should you choose a different value for each magnitude?
   * PC4: Is Q constant during the two weeks?
   * PC5: How do you estimate Q?
   * PC6: What are its values? Does it depend on the patient? PC7: It is interesting to include all the regions of the EGA, not only the region A. Please, include in Table 2, or another table, the results for regions A to E. PC8: To fully understand the results, you should provide the total number of glucose data (N), and present empirical evidence of the consistency of the method's results for the different patients in order to compare if the performance of the method has differences depending on the patient’s characteristics. PC9: Regarding the evaluation metrics in Table 2, I suppose that you have calculated them per patient, since you have personalized each model for each patient, and, then, averaged the metric for all the patients. E.g., you have calculated the RMSE for patient 1, patient 2, and so on, and then you have averaged the RMSE for the ten patients in order to obtain the mean +- std that appears in the table. If so, the Standard Error (SE) is SE = STD / SQRT(10) and the confidence interval for the difference between two techniques (let us say, ARX and PM) is calculated as

                    (mean(ARX)- mean(PM)) +- 2.262 * SQRT( SE(ARX)^2 + SE(PM)^2)

where 2.262 is the critical value for critical significance levels \alpha =0.05 and degree of freedom 9.

Substituting in the previous equation the values for PH=30, I obtain

20.74 - 20.24 +- 2.262 * SQRT( (2.65/SQRT(10))^2 + 2.84/SQRT(10))^2 ) = (-2.28, 3.28)

that contains the zero. So you cannot reject the null hypothesis (that both means are the same). However, the T subscript in the table for the cells of the ARX and PM methods under the PH=30, RMSE columns state that there is statistical significance in the difference between these two methods in contradiction with my previous calculus. My hypothesis is that I must have misunderstood the meaning of the T subscript. To avoid it, you should explain how you have calculated the averaged values and the meaning of the subscripts T, *, ** in the table.

PC10: I am a bit surprised that your method performs better for the real cohort that for the virtual population in which the patients are also modeled using techniques similar to those used in your paper. Please, provide your comments about this fact.

Round 2

Reviewer 1 Report

The authors addressed all my points.
Just few minor things:

page 5: there is a missing - in the parenthesis "(Equations 6 7)" page 5 line 127: thne instead of then

Author Response

We thank the Reviewer for the thorough review and for pointing out these typos.

page 5: there is a missing - in the parenthesis “(Equations 6 7)”

“(Equations 6 7)” has been changed to “(Equations 6-7)”.  

page 5 line 127: thne instead of then

This typo has been corrected.

Reviewer 2 Report

The authors have addressed all my comments. Nevertheless, there are still some minor typos that should be fixed:

page 5: Equations (5 6) -> Equations (5-6)

line 127: Thne

line 271: Out -> Our

Missing pages in several references. E.g., [7], [12], ...

Author Response

We thank the Reviewer for the thorough review and for pointing out these typos.  

page 5: Equations (5 6) −> Equations (5-6)

“-” has been added between the equation numbers.

line 127: thne

This typo has been corrected.   

line 271: Out −> Our

This typo has been corrected.  

Missing pages in several references. E.g., [7], [12], ...

We have updated the references [7], [8], [12], [33], and [43] by adding the corresponding missing pages.